# Size-Exclusion Chromatography Combined with Ultrafiltration Efficiently Isolates Extracellular Vesicles from Human Blood Samples in Health and Disease

**DOI:** 10.3390/ijms24043663

**Published:** 2023-02-11

**Authors:** Chiara Franco, Anna Ghirardello, Loris Bertazza, Michela Gasparotto, Elisabetta Zanatta, Luca Iaccarino, Hadi Valadi, Andrea Doria, Mariele Gatto

**Affiliations:** 1Unit of Rheumatology, Department of Medicine (DIMED), University of Padova, 35128 Padova, Italy; 2Unit of Endocrinology, Department of Medicine (DIMED), University of Padova, 35128 Padova, Italy; 3Department of Rheumatology and Inflammation Research, Institute of Medicine, Sahlgrenska Academy, University of Gothenburg, Guldhedsgatan 10A, 413 46 Gothenburg, Sweden

**Keywords:** extracellular vesicles, size-exclusion chromatography, ultrafiltration, idiopathic inflammatory myopathies

## Abstract

There is still a need for an efficient method for the isolation of extracellular vesicles (EVs) from human blood that provides a reliable yield with acceptable purity. Blood is a source of circulating EVs, but soluble proteins and lipoproteins hamper their concentration, isolation, and detection. This study aims to investigate the efficiency of EV isolation and characterization methods not defined as “gold standard”. EVs were isolated from human platelet-free plasma (PFP) of patients and healthy donors through size-exclusion chromatography (SEC) combined with ultrafiltration (UF). Then, EVs were characterized using transmission electron microscopy (TEM), imaging flow cytometry (IFC), and nanoparticle tracking analysis (NTA). TEM images showed intact and roundish nanoparticles in pure samples. IFC analysis detected a prevalence of CD63+ EVs compared to CD9+, CD81+, and CD11c+ EVs. NTA confirmed the presence of small EVs with a concentration of ~10^10^ EVs/mL that were comparable when stratifying the subjects by baseline demographics; conversely, concentration differed according to the health status across healthy donors and patients affected with autoimmune diseases (130 subjects in total, with 65 healthy donors and 65 idiopathic inflammatory myopathy (IIM) patients). Altogether, our data show that a combined EV isolation method, i.e., SEC followed by UF, is a reliable approach to isolate intact EVs with a significant yield from complex fluids, which might characterize disease conditions early.

## 1. Introduction

Extracellular vesicles (EVs) are a family of nanoparticles delimited by a lipid bilayer that are released from all cell types and cannot replicate [1,2]. EVs are important mediators in cell-to-cell communication which are continuously released into all body fluids and deliver cell-specific cargo and surface biomarkers in an organ-specific manner [3,4]. EVs are subdivided into different subtypes based on size, density, content, and topology [2]. According to the Minimal Information for Studies of Extracellular Vesicles (MISEV) 2018 guidelines, these nanoparticles are generally defined as “extracellular vesicles” (EVs) based on their biochemical composition, such as tetraspanin positive (CD63+/CD81+) EVs, and subdivided into “small EVs” (<200 nm) and “medium/large EVs” (>200 nm) [5] according to their size. It is recognized that EVs are involved in various physiological processes as well as pathological conditions [6] including cancer and autoimmune diseases [7], and their importance as an evolving biomarker within liquid biopsy is generating significant interest in medical research due to the minimally invasive nature of acquisition [8]. Qualitative and quantitative differences in EVs from healthy donors and patients affected with different conditions have been described [9,10], which poses the issue of methodological reliability to allow comparisons between groups. Several methods to isolate EVs from biofluids have been described in the literature. While differential ultracentrifugation (dUC) has been widely used, other charged-based, affinity-based, and size-based methods, including size-exclusion chromatography (SEC) and ultrafiltration (UF) techniques, have emerged as credible alternatives [11,12,13,14]. In 2014, SEC was adapted into a single-step EV isolation from biological fluids. This technique preserves EV integrity, functionality, and characteristics, avoids aggregates, and maintains purity despite the lower yield [14,15]. EV isolation through the SEC-based method from human synovial fluid was shown to efficiently separate proteins that co-purified with high-speed centrifugation [16]. Research conducted on blood plasma from rats showed SEC isolated a higher yield of EVs than UC [17]. Interestingly, a study established a dichotomic SEC method to separate EVs and proteins from fetal bovine serum (FBS), human serum, and FBS-free cell culture supernatants that requires two elutions of loaded samples to acquire EVs and proteins eluates, without the need of multiple fractionation and pooling operations [18]. The functional activity of EVs can depend on the isolation method, thus a combination of isolation techniques may be needed [17] and recommended to achieve an increased purity of EV samples [5,11,19,20]. UF can be used as a complementary technique, e.g., of ultracentrifugation or as a stand-alone technique, although in the latter case, EV preparations are significantly contaminated. A combination of SEC with UF has been effectively utilized to isolate adipocyte EVs from a stromal vascular fraction and adipose tissue-conditioned media [14]. Studies performed on urinary samples [21,22] and cell culture media [23,24] revealed that UF followed by SEC is an efficient method to isolate EVs and it provides high purity, permitting accurate analyses of EV-related biomarkers by -omics technologies [22,23]. Moreover, recent research which employed cell-conditioned medium provided an initial concentration by UF followed by SEC to efficiently separate the components, and then fractions were subjected to another UF step [25]. EV isolation from human plasma by SEC provides enough EV yield for transcriptional analysis, particle size and concentration evaluation, total protein quantification, and the measurement of contaminants [26]. Interestingly, it has been demonstrated that EV isolation from human plasma through SEC followed by UF for RNomics analysis presented more useful reads for both microRNA and mRNA [27]. So far, it has been recommended to isolate EVs from a small amount of plasma to avoid co-enrichments of other plasma molecules [28,29]. Furthermore, sodium citrate appears to be the best anticoagulant during blood collection to minimize in vitro platelet activation and consequent EV release [29]. Then, storing the plasma samples at –80 °C appears to be ideal for maintaining the original EV profile [29].

Another important aspect of the research on EVs is their characterization [19], which has recently included different types of measurement consistent with the ISEV recommendations [11]. Nanoparticle tracking analysis (NTA) is a mainstay for EV counting and sizing [19]. Flow cytometry techniques are beginning to elucidate single-EV biology and single-cell-derived EV characteristics with high throughput. Particularly, imaging flow cytometry (IFC) operates with a lower limit of detection [19], thereby demonstrating high sensitivity to facilitate multi-parametric single EV analysis [13]. Recently, IFC has been proposed for identification, phenotyping, and determination of the concentration of EVs in peripheral blood plasma without prior isolation [30]. The aim of the present study was to investigate the efficiency of EV isolation and characterization methods not yet defined as the “gold standard”. In the present study, we characterized the EVs using TEM, IFC, and NTA to evaluate the efficiency of EV isolation through SEC followed by UF. Moreover, we compared the reliability of EV isolation protocols across healthy donors and patients affected with autoimmune diseases.

## 2. Results

### 2.1. Patient Demographics

The study was conducted on 130 subjects, of which 65 were healthy donors (HDs) and 65 idiopathic inflammatory myopathy (IIM) patients. Demographic features of the subjects enrolled in the study are reported in Table 1.

### 2.2. Significantly Reduced Protein Concentration in Isolated SEC EVs Compared to PFP Samples

The protein concentration measured was significantly reduced in EV fractions collected after the SEC step compared to the respective PFP samples (mean ± SD 0.069 ± 0.061 mg/mL vs. 52.50 ± 4.466 mg/mL, *p* < 0.0001) (Figure 1). Similar data were obtained when separately analyzing HDs (0.072 ± 0.069 vs. 52.30 ± 4.419 mg/mL, *p* < 0.0001) and IIM patients (0.065 ± 0.053 vs. 52.71 ± 4.537 mg/mL, *p* < 0.0001). The protein concentration in PFP samples in healthy and disease groups is not significantly different (52.30 ± 4.419 vs. 52.71 ± 4.537 mg/mL, *p* = 0.5769), nor the protein concentration in EV fractions (0.07282 ± 0.06931 vs. 0.06585 ± 0.05385, *p* = 0.4500) (Appendix A).

### 2.3. Observation of EV Morphology Using TEM

Transmission electron microscopy (TEM) was used to characterize the morphology of the isolated EVs. The TEM images of the isolated EVs (n = 9) showed intact and roundish nanoparticles with a heterogeneous size (diameter range: 30–200 nm) and a prevalence of small EVs in both HDs and IIM patients (Figure 2A,B). Fresh and frozen/thawed (4 or 8 months) EV samples did not display morphological differences (Figure 2C).

### 2.4. IFC Characterization Determines the Presence of EV Markers

The isolated EVs were further characterized for common EV surface protein markers, i.e., tetraspanins CD63, CD81, CD9, and integrin CD11c. A total of 60 EV samples were analyzed to ensure the presence/absence and expression levels of these markers in the isolated EVs. An analysis of the single markers (one marker at a time was analyzed) shows that these EVs are positive for tetraspanins CD63, CD81, CD9, and integrin CD11c (Figure 3). However, the CD63 protein is most expressed in these EVs compared to the other surface protein markers that were studied, i.e., CD81, CD9, and CD11c. Next, our results show that CD81 and CD9 are also highly represented in these EVs after CD63. Mean concentrations of double-positive EVs showed higher values of CD63+/CD11c+ EVs compared with CD63+/CD9+, CD9+/CD11c+, and CD63+/CD81+ particles, while the co-presence of all tetraspanins (CD63+/CD81+/CD9+) reported the lowest concentration (Figure 3).

By comparing the concentration of differently combined tetraspanin-positive and integrin-positive EVs by demographic features and disease, no significant differences (*p* = ns) could be detected (Appendix A).

### 2.5. EV Measurement via NTA

NTA measurements of EV concentration, mean size, and mode size are reported in Appendix A and Figure 4. A total of 130 EV samples were analyzed. The results show that these EVs have a mean concentration of 1.51 × 10^10^ EVs/mL ± 1.06 × 10^10^ SD, mean size of 201.6 nm ± 19.04 SD, and mode size of 153.4 nm ± 18.40 SD.

By comparing EV features across different subgroups, IIM patients displayed a higher mean EV concentration (1.71 × 10^10^ ± 1.29 × 10^10^ vs. 1.31 × 10^10^ ± 7.17 × 10^9^, *p* = 0.0306) compared to HDs (Figure 5). By contrast, no significant differences were shown in the pooled cohort according to gender (males vs. females: 1.46 × 10^10^ ± 9.28 × 10^9^ vs. 1.54 × 10^10^ ± 1.12 × 10^10^, *p* = 0.6737) and no correlation was found between EV concentration and age (Pearson’s r = 0.1369, *p* = 0.1203).

## 3. Discussion

The EV research area is constantly and rapidly evolving with increased interest in their use as biomarkers both in diagnosis and treatment, given their accessibility in biological fluids [31]. However, EV isolation is still evolving due to the need for methodological approaches ensuring a reliability to maintain the original profile of the EVs and the purity of the samples [5,19,20,32]. Hence, it is paramount to establish a validated methodology to efficiently separate and characterize EVs. In this study, we have obtained an efficient separation of EVs through SEC followed by UF, as confirmed by the several characterization techniques that demonstrated intact EVs in pure samples. According to the International Society on Thrombosis and Hemostasis guidelines on plasma EVs, we employed PFP samples as the starting material to avoid the presence of platelet-derived EVs [33,34]. Despite ultracentrifugation remaining the gold standard to isolate EVs, it has been demonstrated that SEC methods can preserve vesicle structure, integrity, and biological activity [32]. However, conflicting data exist concerning the purification of EVs from high-density lipoproteins (HDL) and proteins from plasma and other macromolecules, as during SEC, the persistence of low-density proteins and HDL can occur [12,27,35]. Hence, it has been suggested to combine SEC with other techniques, such as UF [5,12]. In keeping with former evidence, our EV isolation method suggests an efficient EV yield in terms of EV concentration [36] and the separation of vesicular and protein fractions. We enriched the isolated EVs through UF and then adjusted the samples at the initial PFP volume (500 µL) to measure plasma EV concentration via NTA, which led to a determination of the real concentration of circulating EVs.

EV characterization performed via TEM is a useful method to discriminate single EVs, EV aggregates, and non-EV particles similar in size, despite some degree of operator-dependent selection [37]. Through TEM analysis of isolated EV fractions, the EV integrity was testified by a typical “cup-shape” EV morphology in pure samples, thus demonstrating EV integrity following SEC combined with UF techniques. Our samples of isolated particles were enriched in small EVs.

Thus, we performed immune-characterization through IFC, a technique that allows the detection of small particles and the characterization of specific EV phenotypes [38]. In fact, conversely to conventional flow cytometers with a low detection limit that exclude smaller EVs from the analysis [35], the Imagestream^X^ MKII instrument combines increased fluorescence sensitivity, low background, and powerful data analysis [39]. To solve the requirement for fluorescent markers because small EVs are below the size of the brightfield camera resolution and have low scatter levels [39], we verified the presence of constitutive EVs markers using fluorescent-labelled antibodies, thereby confirming that isolated nanoparticles were in fact EVs according to their biochemical surface composition. In our study, IFC analysis confirmed the presence of plasmatic tetraspanin-positive EVs, the majority being CD63+ EVs compared to CD9+, CD81+, and integrin CD11c+ EVs. This is in keeping with previous results showing that the qEV SEC column isolation approach from plasma samples allows the recovery of the majority of CD63+ and CD9+ EVs [35]. Conversely, a study conducted by Veziroglu et al. reported that CD9 and CD81 were consistent EV markers, while CD63 expression depended on the experimental parameters [19]. In fact, other researchers demonstrated the prevalence of CD9+ EVs in blood with differences in concentration according to anti-coagulants, plasma, or serum biofluid type, and isolation method, while CD81+ EVs were the rarest tetraspanin in both plasma and serum, and CD63+ EVs were enriched in serum. However, the researchers affirmed that CD63 is present in multiple subpopulations of EVs, and both CD63+ and CD81+ EVs were less affected by the type of anticoagulant used [40]. In addition, it has been demonstrated that the tetraspanin composition of EVs differs according to the collected SEC fractions [25], as well as to their cellular origin, which might influence the selection of target cells and the mechanism of EV uptake [41]. Our results show that the co-distribution of tetraspanin markers on EVs is mainly characterized by CD63/CD9 expression, followed by CD63/CD81, and CD81/CD9. In addition, the expression of the integrin CD11c associated with CD63 appears the most abundant EV subpopulation in plasma and CD81/CD11c is the least represented co-distribution of double-positive markers EVs. Accordingly, a recent study by Karimi and colleagues assessed the non-mutual existence of CD63 and CD81 markers. Their results showed that CD63/CD81+ EVs were very few in both EDTA plasma and serum, CD63/CD9+ EVs were more abundant in serum than in plasma, whereas CD81/CD9+ EVs were the least common EV type. These observations suggest that blood contains diverse subpopulations of EVs that express tetraspanins in a different distribution, whose detection can be influenced by several experimental and intrinsic factors including the cellular origin (e.g., CD81 is expressed on B and T lymphocytes and not on erythroid cells or platelet-derived EVs [40]). It should be noted that, unlike our study, the study of Karimi et al. did not include sodium citrate as an anticoagulant as well as the combination of SEC with UF [40]. In accordance with Botha and colleagues, we speculate that in clinical studies with expected small differences in EV concentrations between groups, platforms with a higher reproducibility would be preferable to spot slight differences [38], although IFC is fundamental to the biochemical characterization of EVs.

EV quantification introduced several challenges, predominantly due to their small size and the employment of diverse technologies [35]. NTA technology is the gold standard used to determine EV concentration and size distribution. However, one of the limitations of NTA is that it is not capable of distinguishing between EVs and similar-sized contaminants from blood samples, which could lead to an overestimation of the EV concentration [36,42]. In keeping with former reports on human blood plasma from healthy subjects [36], the data obtained from NTA measurements in the present study reported EV concentration values of ~10^10^ EVs/mL, confirming the efficiency of our EV isolation methodology. The diameter of isolated EVs across our samples is registered to be consistently lower than 200 nm, confirming the predominance of small EVs isolated using qEV original^®^/70 nm SEC columns among circulating EVs in plasma samples. In fact, former evidence reported the vast majority of circulating EVs to be smaller than 200 nm in diameter [19], in addition to the observation that smaller EVs are detected when isolated by SEC than ultracentrifugation [12]. In addition, the results obtained from NTA highlighted significant differences between groups of patients and HDs, supporting a good sensitivity of this technique. Moreover, the general homogeneity of nanoparticle concentrations and sizes guarantees the reliability of the isolation methods. 

Our study has some limitations which must be considered. We did not directly compare different techniques for the EV isolation. Consequently, the prevalence of small EVs in our samples could be due to a real representation of the circulating particles or to a dimensional selection of the particles during the SEC step, despite the range of size selection of the used Izon columns being 70–1000 nm. Furthermore, the efficiency of particle isolation in terms of EV concentration was proposed by comparing our results with the data in the literature. Moreover, we did not verify the absence of platelets in plasma samples or specific protein contaminants in EV fractions. On the other hand, the main strength of our study consists of the efficacy and reproducibility of the EV isolation through the combined SEC and UF approach, confirmed by the subsequent application of TEM, IFC, and NTA characterization techniques. In addition, isolated EVs were proven comparable in the pooled cohort in terms of concentration, size, and immunophenotyping; on the other hand, we documented a compelling difference in EV concentration when comparing health and disease, submitting evidence for our isolation technique to reliably separate the total circulating EVs from HDs and other phenotypes.

## 4. Materials and Methods

Unselected healthy donors (HDs) and patients affected with idiopathic inflammatory myopathies (IIM), classified according to Bohan and Peter’s criteria [43,44] and followed up at the Unit of Rheumatology of Padova University Hospital, were included in the study. Subjects classified as HDs were not on any chronic medication at the time of enrolment. Demographics, clinical, laboratory, and treatment variables were recorded. EV isolation and processing were conducted with the same methodology among HDs and IIM patients and differences in retrieval rate, concentration, and size of EVs were tested according to demographic features and health status. For the purpose of the present study and due to sample size limitations, IIM patients were considered as a single disease group. This study was conducted in accordance with the Declaration of Helsinki and approved by the local ethics committee. All participants gave written informed consent.

### 4.1. Human Platelet-Free Plasma Collection

Peripheral blood venous samples (6 mL) were collected from non-fasting subjects in sodium-citrate tubes and platelet-free plasma (PFP) was prepared within 1 h. Briefly, the blood samples in Vacutainer tubes were centrifuged at 1500× *g* for 20 min to separate plasma from cells, and then plasma supernatants were transferred into clean tubes and centrifuged twice at 3000× *g* for 15 min to remove platelets. PFP samples were aliquoted leaving a residue of plasma above the pellet area and stored at −80 °C. A similar protocol to obtain PFP samples has been introduced by the International Society on Thrombosis and Hemostasis [45] and it has been experimentally validated to efficiently remove platelets for EV studies [34].

### 4.2. Size-Exclusion Chromatography (SEC)

PFP samples were defrosted at room temperature (RT) to perform the EV isolation steps. The SEC procedure was performed using qEV original^®^/70 nm columns (Izon Science), following the manufacturer’s instructions. Briefly, the Izon column was removed at +4 °C and the storage solution was allowed to run. The column was equilibrated with 10 mL of Phosphate Buffered Saline (PBS) (pH 7.4; ThermoFisher Scientific, Waltham, MA, USA) filtered through a 0.22 µm filter unit (Millex—GP; Merck Millipore) (fPBS).Then, 0.5 mL of the PFP sample was added on the top, and once absorbed into the column, fPBS was added to keep it from drying out, subsequently subdividing the eluate into 25 fractions, each one of 0.5 mL. Fractions 1–6 (3 mL) were the void volume which was disposed of. Fractions 7–10 (2 mL) containing the vesicular fraction were collected for further processing, and fractions 11–25 (7.5 mL) containing the protein fraction were eliminated. 

### 4.3. Ultrafiltration (UF)

The EV fractions collected by SEC were enriched through ultrafiltration (UF) using an Amicon^®^ Ultra-4 mL, 100 KDa centrifugal filter unit (Merck Millipore) which does not concentrate small molecules. Each filter was sterilized by centrifugation with 1 mL of ethanol 70% at 2800× *g* for 1 min. The ethanol residues were removed by inversion and then by centrifugation with 2 mL of fPBS at 2800× *g* for 2 min. EV SEC fractions (2 mL) + 1 mL of fPBS were added above the previously ethanol-sterilized filter and centrifuged at 4000× *g* for 10 min, according to the manufacturer’s instructions, to collect the samples held on the filter containing particles of ≥100 KDa, including EVs, and remove particles of <100 KDa that pass through the filter (e.g., albumin). The collected samples were adjusted to a 0.5 mL volume by adding fPBS, aliquoted, and frozen at −80 °C.

### 4.4. Nanodrop Spectrophotometer

The protein concentration of both PFP and respective SEC-eluted EV samples was quantified using a NanoDrop 2000C^®^ Spectrophotometer (Thermo Fisher Scientific) using the function “Protein A280”.

### 4.5. Transmission Electron Microscopy (TEM)

One drop of EV samples (about 25 µL) was placed onto a 400 mesh holey film grid. After staining with 2% uranyl acetate (for 2 min), the sample was observed with a Tecnai G2 (FEI) transmission electron microscope operating at 100 kV. Images were captured by a Veleta (Olympus Soft Imaging System) digital camera.

### 4.6. Imaging Flow Cytometry (IFC)

EV sample acquisitions of fluorescent signals were performed using an Amnis ImageStream^X^ MkII (ISx; EMD Millipore, Seattle, WA, USA) instrument. EV samples were diluted in fPBS and then incubated for 30 min in the dark at RT with the following fluorophore-conjugated antibodies previously centrifuged at 20,000× *g* for 10 min at 10 °C to prevent the formation of antibody–antibody aggregates: anti-human CD63 (Brilliant Violet 421; Biolegend, clone H5C6; RRID:AB_2687004) in channel 1 (435–505 nm filter), anti-human CD81 (FITC; Biolegend, clone 5A6; RRID:AB_10642825) in channel 2 (505–560 nm filter), anti-human CD9 (Alexa Fluor 647; Exbio, clone MEM-61; RRID:AB_10733480) in channel 5 (642–745 nm filter), and anti-human CD11c (PE; Biolegend, clone 3.9; RRID:AB_314176) in channel 3 (560–595 nm filter). Channel 4 (595–642 nm filter) was set to brightfield (BF) and channel 6 (745–780 nm filter) to side-scatter (SSC). The acquisition settings were applied for a fixed time of 2 min with low-speed fluidics, and all used a laser (405 nm, 488 nm, 642 nm, 785 nm) at maximum power to ensure maximal sensitivity, with magnification at 60×, a core size of 7 µm, numerical aperture of 0.9, DOF of 2.5 µm, and the “Remove Beads” option unchecked to include Speed Beads (1000 nm) in the analyses. Technical controls included: labelled EVs with 0.1% Triton X-100 (Merck), unlabeled EVs, single antibody-labelled EVs, and mixed samples of single antibody-labelled EVs, fPBS with the addition of fluorophore-conjugated antibodies, and fPBS. An improved masking setting (NMC, not in the MC mask) was used to optimize the SSC resolution and efficacy of the detection of dim fluorescent single EVs compared to the “masks combined” (MC) standard setting. An EV gating strategy was performed by applying the “Intensity_MC Or NMC” of SSC (channel 6) versus the “Normalized Frequency” in a histogram to identify the low SSC area limited to the area under the Speed Beads. The resulting dot-plot graphs provided the “Intensity of SSC channel” in the Y axis and “Intensity_MC Or NMC” for each channel (2, 3, and 5) in the X axis to detect the fluorescent-positive particles (Appendix A). Upon each start-up, the instrument calibration tool ASSIST^®^ was utilised to optimize performance and consistency. Samples were acquired using INSPIRE^®^ software. Data analyses and spectral compensation matrices were performed using IDEAS^®^ software (version 6.3). The advanced ISx fluidic control coupled with the continuously running Speed Beads enabled particle enumeration using the “objects per mL” feature within the IDEAS^®^ software.

### 4.7. Nanoparticle Tracking Analysis (NTA)

EV quantification and size were measured in samples diluted in fPBS to the concentration range of 10^6^–10^8^ particles/mL using the NanoSight NS300 (Malvern Panalytical) instrument, as specified by the manufacturer, for the optimal assessment of particle concentration and size distribution. The ideal detection threshold was determined to include particles with the restriction concentrations of 20–120 particles per frame, while indistinct particles were limited to 5 per frame. According to the manufacturer’s manual, the camera level was increased to visibly distinguish all particles not exceeding a particle signal saturation over 20% and autofocus was adjusted to avoid indistinct particles. For each sample, particles moving under Brownian motion were recorded on three 1 min videos captured with a 20× magnification. EV concentration and size were calculated using NTA software (version 3.4). To minimize data skewing based on single large particles, the ratio between total valid tracks and total complete tracks was always ≤1:5.

### 4.8. Statistics

Data were expressed as mean and standard deviation (SD) for continuous variables while categorical data were expressed as numbers and percentages. A Shapiro–Wilk test was performed to test normality. Student’s t test was used to compare the differences between parametric variables, and one-way ANOVA with Bonferroni’s correction was used when more than two groups were considered. Pearson’s correlation was used to evaluate the strength and direction of the correlation. A two-tailed *p*-value (*p*) ≤ 0.05 was considered statistically significant. Data were analyzed using SPSS software version 20.0 (SPSS, Chicago, IL, USA) and GraphPad Prism^®^ version 9.

## 5. Conclusions

Our results show that the combined EV isolation protocol consisting of SEC followed by UF, and the EV characterization by both traditional and novel techniques, including TEM, IFC, and NTA, is a reliable approach to obtain and characterize pure EV fractions from complex human biological fluids such as plasma, with preserved morphological integrity. The methodology described herein appeared reproducible without significant differences concerning EV features (morphology, size, and surface markers) between healthy and disease groups In addition, our data support a consistent discrimination in terms of EV concentration between HDs and diseased subjects. Thus, we suggest these methods to be suitable for investigating EV features, the biocompatibility of EVs as drugs carriers, EV cargo, and as potential disease biomarkers to be further implemented in clinical practice.

## Figures and Tables

**Figure 1 ijms-24-03663-f001:**
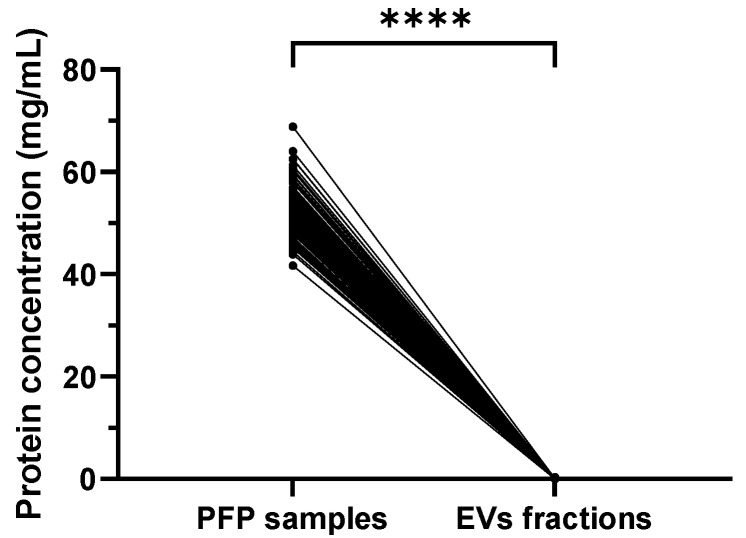
Graph representing the protein concentration before SEC isolation (PFP samples) and after SEC isolation (EV fractions) (n = 130); **** *p* < 0.0001. The total protein concentration was drastically reduced after isolation of EVs using the SEC method.

**Figure 2 ijms-24-03663-f002:**
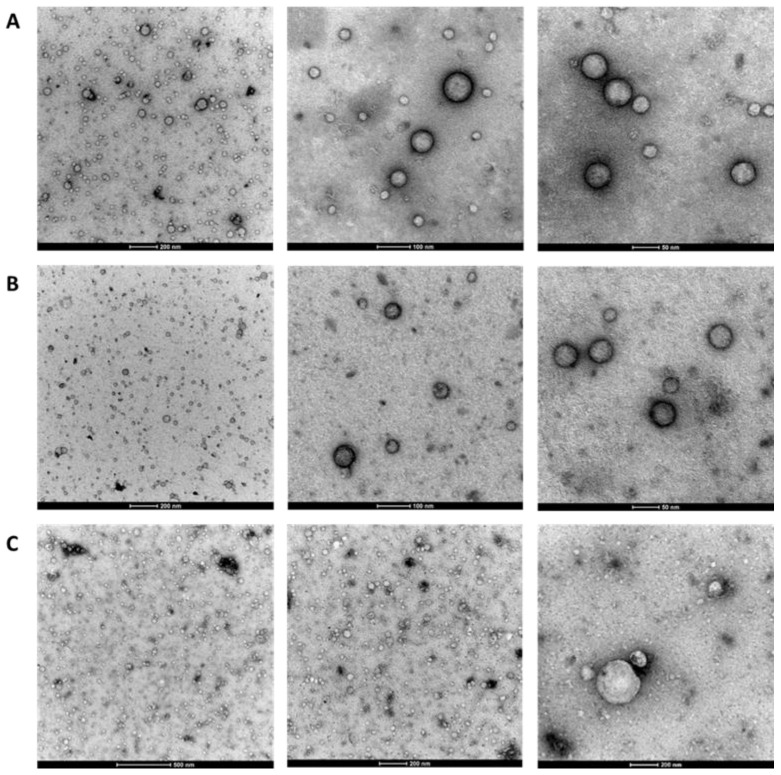
Representative TEM images of the isolated EVs. IIM patient (n = 1) (**A**), HD (n = 1) fresh samples (dilution 1:10) (**B**), and IIM patient (n = 1) after freeze (8 months) and thawed sample cycle (undiluted) captured at high magnification (**C**) (scale bar: 50 nm; 100 nm; 200; and 500 nm).

**Figure 3 ijms-24-03663-f003:**
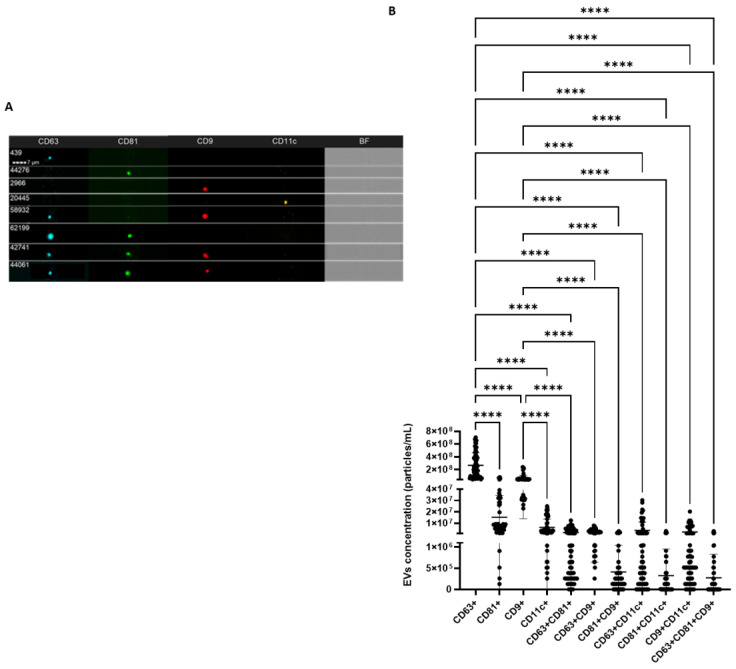
(**A**) shows representative captured particles with a positive signal to the tetraspanins CD63, CD81, CD9, and the integrin CD11c. (**B**) Graphs reporting the EV concentration (mean ± SD) of single- double- and triple- positive EVs to the surface markers (n = 60). BF: Brightfield; **** *p* < 0.0001.

**Figure 4 ijms-24-03663-f004:**
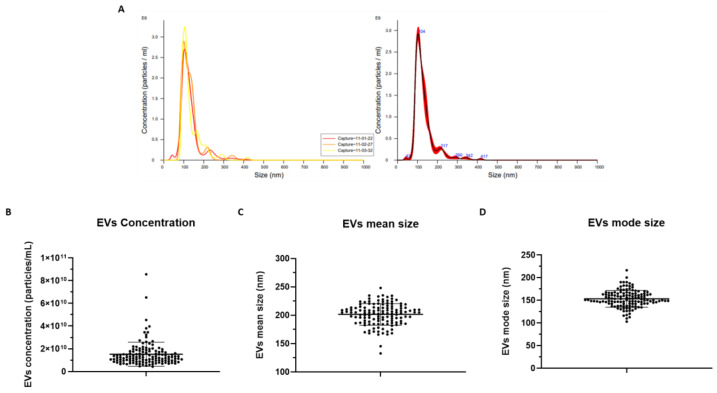
A representative graph of NTA measurement through NanoSight NS300 (Malvern Panalytical) obtained by NTA Software version 3.4, reporting the values of particle concentration in relation to their size. On the left, the three curves show the data captured in three 1 min videos for the same sample. On the right, the merged data (**A**). Graphs representing the EV mean concentration ([EVs/mL] ± SD) (**B**), EV mean size (nm ± SD) (**C**), and EV mode size (nm ± SD) (**D**) (n = 130).

**Figure 5 ijms-24-03663-f005:**
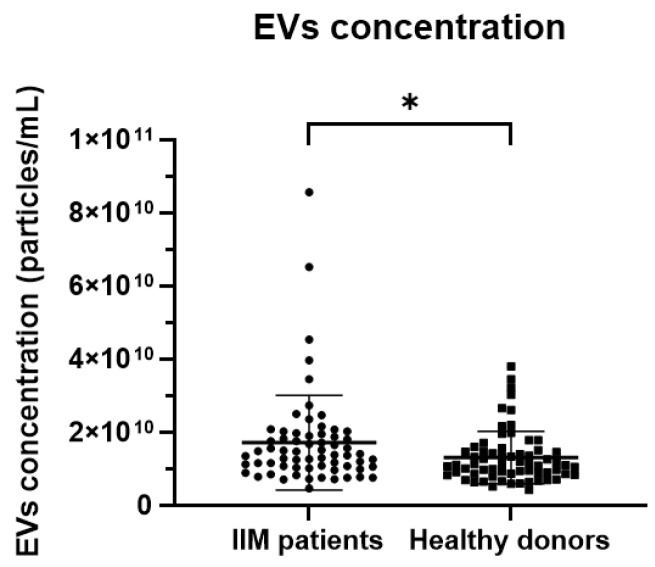
Graphs representing the EV mean concentration ([EVs/mL] ± SD) in IIM patients (n = 65) and HDs (n = 65); * *p* < 0.05.

**Table 1 ijms-24-03663-t001:** Demographic features of the subjects enrolled in the study.

Demographic Features of Recruited Subjects	
**Total subjects**, n data	130
Age at blood collection time (years; mean ± SD)	55.45 ± 16.78
Females, n (%)	86 (66.1)
**Healthy donors**, n (%)	65 (50)
Females, n (%)	43 (33.07)
Age at blood collection time (years; mean ± SD)	52.28 ± 17.22
**IIM patients**, n (%)	65 (50)
Females, n (%)	43 (33.07)
Age at blood collection time (years; mean ± SD)	60.51 ± 12.75

IIM: idiopathic inflammatory myopathies.

## Data Availability

Data is contained within the article or Appendix A.

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
