# Peer review of "Size-Exclusion Chromatography Combined with Ultrafiltration Efficiently Isolates Extracellular Vesicles from Human Blood Samples in Health and Disease"

_ijms, 2023, doi:10.3390/ijms24043663_

Round 1
Reviewer 1 Report
This paper intends to characterize the EV fraction isolated from “platelet free plasma” by SEC followed by ultrafiltration. It is an interesting piece of work. However, the novelty is limited as several other papers (not mentioned in the bibliography) have already been published in the field (e.g. Yang et al https://doi.org/10.1186/s12967-021-02775-9 or Tkach et al https://doi.org/10.1073/pnas.2107394119). Moreover, several affirmations are done without any evidence or proper bibliographical reference. The affirmation in the conclusion that “we propose a newly-conceived EVs isolation protocol, consisting of SEC followed by UF” is thus loose, as the methodology has been published elsewhere.
Thus, points to be clarified are:
1.- If samples are supposed to be platelet free it needs to be shown that EVs are negative for platelet markers.
2.- This is a methodological paper. Thus, the vesicle concentration and protein content of the SEC fractions need to be shown at least in the supplementary material. How did the authors know which fractions should be discarded in each case? Did the SEC fractions vary between healthy and disease? Were differences in protein concentration/protein EV load observed between groups?
Why SEC fractions were enriched using 100 KDa centrifugal filter units (see Tkach et al, https://doi.org/10.1073/pnas.2107394119)? This means that contaminating proteins up to 100 KDa will be in the pellet. If “the removal of the majority of high-density lipoproteins and proteins from plasma and other macromolecules is guaranteed, contamination by other lipoproteins is not excluded”. The content of these macromolecules in the fractions need to be shown to support the pooling of the mentioned fractions…. The co-distribution of surface markers (albeit using different EV isolation methods) has been reported elsewhere and should be discussed (e.g. Donzelli et al, DOI: 10.1002/jev2.12143)
Author Response
Response to Reviewer 1 Comments
This paper intends to characterize the EV fraction isolated from “platelet free plasma” by SEC followed by ultrafiltration. It is an interesting piece of work. However, the novelty is limited as several other papers (not mentioned in the bibliography) have already been published in the field (e.g. Yang et al https://doi.org/10.1186/s12967-021-02775-9 or Tkach et al https://doi.org/10.1073/pnas.2107394119). Moreover, several affirmations are done without any evidence or proper bibliographical reference. The affirmation in the conclusion that “we propose a newly-conceived EVs isolation protocol, consisting of SEC followed by UF” is thus loose, as the methodology has been published elsewhere.
We thank the reviewer for the careful reading of our paper. According to the reviewer’s suggestion, we have implemented the background with additional updated references, and we have elaborated on SEC and UF EVs isolation methods. Then, we modified the conclusion specifying that this paper aims to suggest both the EVs isolation and characterization methods in combination as valid techniques to study EVs from human plasma (pages 10-11, lines 180-391): “Conclusions: Our results show that the combined EVs isolation protocol consisting of SEC followed by UF and the EVs characterization by both traditional and novel techniques TEM, IFC, and NTA is a reliable approach to obtain and characterized pure EVs fractions from complex human biological fluids such as plasma, with preserved morphological integrity. The methodology described herein appeared reproducible without significant differences between health and disease concerning EVs features (morphology, size, and surface markers). Besides, our data support consistent discrimination in terms of EVs concentration between HDs and diseased subjects. Thus, we suggest these methods to be suitable for investigating EVs features, EVs biocompatibility as drugs carriers, EVs cargo, and as potential disease biomarkers to be further implemented in clinical practice.”
1.- If samples are supposed to be platelet free it needs to be shown that EVs are negative for platelet markers.
Response 1. Thanks for raising this point which we need to clarify. We state that the plasma obtained is platelets-free as our protocol follows the recommendations described in “qEV Application Note” – “Materials and Methods” section provided by Izon Science: “To isolate EVs from plasma, draw 4 mL of human venous blood from a patient into a BD Vacutainer® tube containing sodium citrate as an anticoagulant. Mix and centrifuge (1,500 x g, 20 mins) the Vacutainer tube to separate the plasma from the cells. Transfer the plasma supernatant to a clean test tube and centrifuge (3,000 x g, 15 mins). Transfer the plasma supernatant to a clean test tube and centrifuge again. Transfer aliquots of platelet-free plasma (PFP) supernatant to several 1.5 mL microcentrifuge tubes leaving a residue of plasma above the pellet area”.
We have added a more detailed description of plasma processing in the Methods section (pages 8-9, lines 284-293) to make this point clear for the reader, specifying: “A similar protocol to obtain PFP samples has been introduced by the International Society on Thrombosis and Haemostasis and it has been experimentally validated to efficiently remove platelets for EVs studies.” (Rikkert LG. et al., doi: 10.1080/09537104.2020.1779924 and Lacroix R. et al., doi: 10.1111/jth.12207).
2.- This is a methodological paper. Thus, the vesicle concentration and protein content of the SEC fractions need to be shown at least in the supplementary material. How did the authors know which fractions should be discarded in each case? Did the SEC fractions vary between healthy and disease? Were differences in protein concentration/protein EV load observed between groups? Why SEC fractions were enriched using 100 KDa centrifugal filter units (see Tkach et al, https://doi.org/10.1073/pnas.2107394119)? This means that contaminating proteins up to 100 KDa will be in the pellet.
Response 2. We added Supplementary Table S1 “Protein concentrations in PFP samples and EVs fractions” demonstrating the protein concentration [mg/mL] before (PFP samples) and after (EVs fractions) SEC isolation in both patients and healthy donors. Moreover, Supplementary Table S4 reports the EVs concentration in both patients and healthy donors.
SEC fractions that were discarded and those collected are specified in the Izon datasheet for qEV/original 70 nm columns “qEVORIGINAL USER MANUAL”. Referring to 500 µL of plasma starting material, in paragraphs “3.6 qEVoriginal EV Elution Profile” and “3.7 qEVoriginal Sample Input Volume Effects and Recovery Rates” the manual shows the fractions rich in EVs and the fractions rich in proteins (https://www.schaefer-tec.it/sites/default/files/qEVoriginal_Technical_Note.pdf). As mentioned in the “Material and Methods” section (page 9, line 303), in our study fractions 7-10 containing the vesicular fraction were collected for further processing (these fractions correspond to EVs collected fractions indicated in Tkach et al. ref n. 25), and fractions 11-25 containing the protein fraction were eliminated. Samples of all subjects were processed in the same way. Collected and discarded fractions do not vary between healthy and disease in order not to introduce methodological variables to the results.
The protein concentration in PFP samples between healthy and disease is not significantly different (52.30 ± 4.419 vs. 52.71 ± 4.537 [mg/mL] ± SD, p=0.5769) as well as the protein concentration in EVs fractions (0.07282 ± 0.06931 vs. 0.06585 ± 0.05385 [mg/mL] ± SD, p=0.4500) (page 3, line 112-115). On the same note, no significant differences were found in the comparison between groups for surface markers positivity as reported in Supplementary Table S2.
We enriched EVs fractions using Amicon® 100 KDa centrifugal filter unit because, as reported by our results obtained from the Nanodrop protein quantification (page 3, lines 107-119), the previous SEC step should have already separated most of the protein fraction from EVs fractions subjected to UF. We used this filter unit collecting the sample held on the filter that allows to retain particles ≥ 100 KDa and remove smaller particles that pass through the filter, including tetraspanins not associated with EVs, such as CD63 (63 KDa) as well as the most abundant protein in plasma, albumin of about 65-70 KDa. The 100 KDa centrifugal filter unit reduces the risk of the filter becoming clogged resulting in insufficient loss of contaminants. Finally, the presence of no-EV molecules ≥ 100 KDa in our samples could be detected with the applied characterization techniques (such as TEM, IFC) compared to less ability to detect smaller molecules.
- If “the removal of the majority of high-density lipoproteins and proteins from plasma and other macromolecules is guaranteed, contamination by other lipoproteins is not excluded”. The content of these macromolecules in the fractions need to be shown to support the pooling of the mentioned fractions.
We have modified the sentence specifying that this concern is reported in the literature, therefore supporting the need for a coupled (SEC+UF) isolation method (page 7, lines 184-187). In our study, we observed the significantly reduced protein concentration in SEC-EV fractions compared to the starting material suggesting a separation of EVs from protein content (page 3, lines 107-115 and Supplementary Table S1).
- The co-distribution of surface markers (albeit using different EV isolation methods) has been reported elsewhere and should be discussed (e.g. Donzelli et al, DOI: 10.1002/jev2.12143).
We widened the Discussion in our manuscript with attention to EVs from blood samples (page 7, lines 222-236).
Reviewer 2 Report
In this work, Franco et al combine already described with alternative protocols for the isolation of characterization of extracellular vesicles of human blood samples. The main novelty, according to the authors’ statement, is the application of size-exclusion chromatography for the isolation combined with ultrafiltration, followed by several characterization procedures, such as imaging flow cytometry and nanoparticle tracking analysis.
The work does not provide great novelty to the field, but it is worthy to know the claimed advantages of size-exclusion chromatography in this kind of samples, which has been largely described for other types of extracellular vesicles, either from eukaryotes or prokaryotes. The authors apply the protocol to a cohort of human samples and they do not find significant differences between the control group (healthy donors) and the disease group.
In general, the manuscript is well written. Some minor points should be addressed before the final acceptance of the work:
-First, the template corresponds to other journal (Acoustics). Please change it and check if the structure fits with the journal to which the manuscript has been submitted (i.e. IJMS).
-Figure 3 is too small. Text within the figure cannot be read. Please enlarge it.
-Line 140: please remove the piece of text in Italian.
-The conclusions are not supported by the results, as no differences in EV protein concentration nor surface markers were obtained between healthy donors and disease patient samples. However, the authors claim in the conclusions that the protocol proposed in this manuscript can serve for differentiating between patient cohorts. Please rewrite the conclusions and/or address this issue in other way.
Author Response
Response to Reviewer 2 Comments
In this work, Franco et al combine already described with alternative protocols for the isolation of characterization of extracellular vesicles of human blood samples. The main novelty, according to the authors’ statement, is the application of size-exclusion chromatography for the isolation combined with ultrafiltration, followed by several characterization procedures, such as imaging flow cytometry and nanoparticle tracking analysis.
The work does not provide great novelty to the field, but it is worthy to know the claimed advantages of size-exclusion chromatography in this kind of samples, which has been largely described for other types of extracellular vesicles, either from eukaryotes or prokaryotes. The authors apply the protocol to a cohort of human samples and they do not find significant differences between the control group (healthy donors) and the disease group.
In general, the manuscript is well written. Some minor points should be addressed before the final acceptance of the work.
We thank the reviewer for the positive feedback and careful suggestions. We have widened the background by quoting more updated references and elaborating on SEC and UF EVs isolation methods. Then, we modified the conclusion specifying that this paper aims to suggest both the EVs isolation and characterization methods in combination as valid techniques to study EVs from human plasma (pages 10-11, lines 180-391): “Conclusions: Our results show that the combined EVs isolation protocol consisting of SEC followed by UF and the EVs characterization by both traditional and novel techniques TEM, IFC, and NTA is a reliable approach to obtain and characterized pure EVs fractions from complex human biological fluids such as plasma, with preserved morphological integrity. The methodology described herein appeared reproducible without significant differences between health and disease concerning EVs features (morphology, size, and surface markers). Besides, our data support consistent discrimination in terms of EVs concentration between HDs and diseased subjects. Thus, we suggest these methods to be suitable for investigating EVs features, EVs biocompatibility as drugs carriers, EVs cargo, and as potential disease biomarkers to be further implemented in clinical practice.”
-First, the template corresponds to other journal (Acoustics). Please change it and check if the structure fits with the journal to which the manuscript has been submitted (i.e. IJMS).
The manuscript has been adapted to IJMS template.
-Figure 3 is too small. Text within the figure cannot be read. Please enlarge it.
Figure 3 has been enlarged.
-Line 140: please remove the piece of text in Italian.
We apologize for the typo. The sentence has been deleted.
-The conclusions are not supported by the results, as no differences in EV protein concentration nor surface markers were obtained between healthy donors and disease patient samples. However, the authors claim in the conclusions that the protocol proposed in this manuscript can serve for differentiating between patient cohorts. Please rewrite the conclusions and/or address this issue in other way.
We modified the conclusions underling that the homogeneity of the results suggests the reliability of the isolation protocol as the starting material and processing are the same for all samples. In this regard, the significantly higher EVs concentration in disease than healthy donors allows to propose NTA as able to detect differences in EV levels with interest in clinical applications. On the other hand, the IFC technique is useful to characterize the EVs in terms of surface markers, but we propose the latter as not adequate to detect small differences in EVs concentrations.
Round 2
Reviewer 1 Report
The major concerns were addressed by the authors